# Fast Training of Diffusion Models with Masked Transformers

**Hongkai Zheng**[*]                                                              *hzzheng@caltech.edu*
*Caltech*

**Weili Nie**[*]                                                                     *wnie@nvidia.com*
*NVIDIA*

**Arash Vahdat**                                                              *avahdat@nvidia.com*
*NVIDIA*

**Anima Anandkumar**                                                       *anima@caltech.edu*
*Caltech*

**Reviewed on OpenReview:** *https://openreview.net/forum?id=vTBjBtGioE*

## Abstract

We propose an efficient approach to train large diffusion models with masked transformers. While masked transformers have been extensively explored for representation learning, their application to generative learning is less explored in the vision domain. Our work is the first to exploit masked training to reduce the training cost of diffusion models significantly. Specifically, we randomly mask out a high proportion (*e.g.*, 50%) of patches in diffused input images during training. For masked training, we introduce an asymmetric encoder-decoder architecture consisting of a transformer encoder that operates only on unmasked patches and a lightweight transformer decoder on full patches. To promote a long-range understanding of full patches, we add an auxiliary task of reconstructing masked patches to the denoising score matching objective that learns the score of unmasked patches. Experiments on ImageNet-256×256 and ImageNet-512×512 show that our approach achieves competitive and even better generative performance than the state-of-the-art Diffusion Transformer (DiT) model, using only around 30% of its original training time. Thus, our method shows a promising way of efficiently training large transformer-based diffusion models without sacrificing the generative performance. Our code is available at https://github.com/Anima-Lab/MaskDiT.

## 1 Introduction

Diffusion models (Sohl-Dickstein et al., 2015; Ho et al., 2020; Song et al., 2021b) have become the most popular class of deep generative models, due to their superior image generation performance (Dhariwal & Nichol, 2021), particularly in synthesizing high-quality and diverse images with text inputs (Ramesh et al., 2022; Rombach et al., 2022; Saharia et al., 2022b; Balaji et al., 2022). Their strong generation performance has enabled many applications, including image-to-image translation (Saharia et al., 2022a; Liu et al., 2023), personalized creation (Ruiz et al., 2022; Gal et al., 2022) and model robustness (Nie et al., 2022; Carlini et al., 2022). Large-scale training of these models is essential for their capabilities of generating realistic images and creative art. However, training these models requires a large amount of computational resources and time, which remains a major bottleneck for further scaling them up. For example, the original stable diffusion (Rombach et al., 2022) was trained on 256 A100 GPUs for more than 24 days. While the training cost can be reduced to 13 days on 256 A100 GPUs with improved infrastructure

---

[*]Equal contribution. Correspondence to hzzheng@caltech.edu and wnie@nvidia.com.

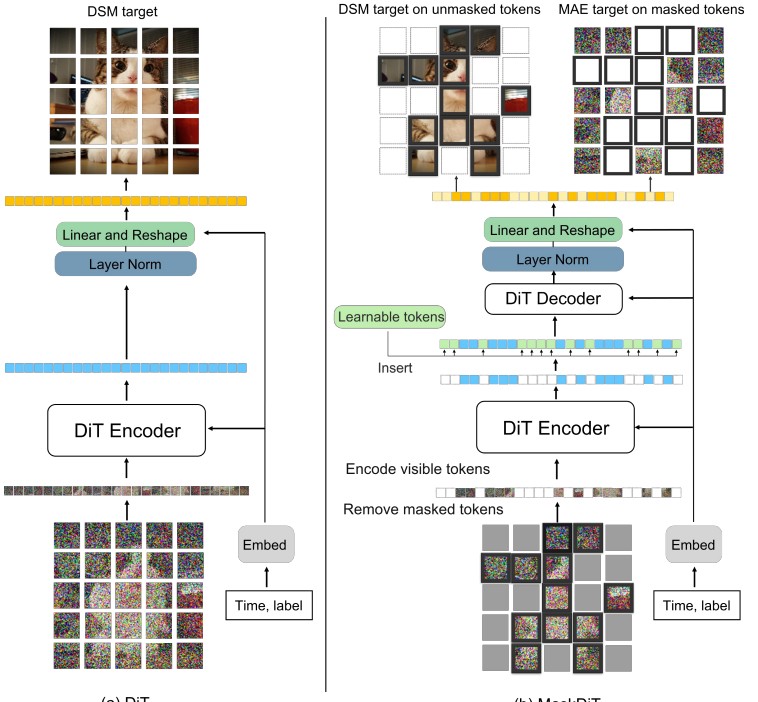

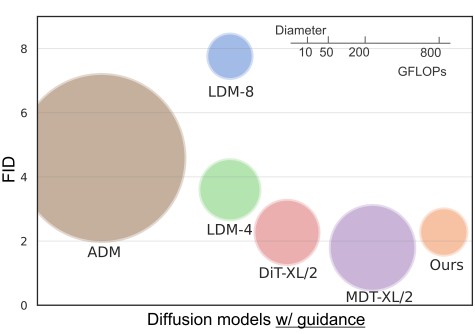

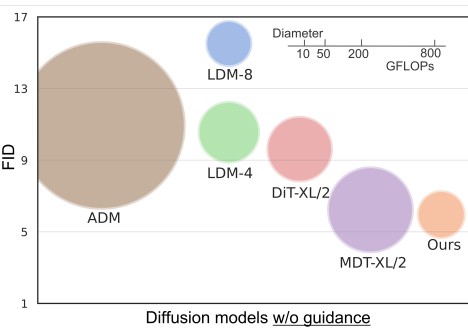

Figure 1: A comparison of our MaskDiT architecture with DiT (Peebles & Xie, 2022). During training, we randomly mask a high proportion of input patches. The encoder operates on unmasked patches, and after adding learnable masked tokens (marked by gray), full patches are processed by a small decoder. The model is trained to predict the score on unmasked patches (DSM target) and reconstruct the masked patches (MAE target). At inference, all the patches are processed for sampling.

Figure 2: Generative performance of the state-of-the-art diffusion models on ImageNet-256×256, in two settings: with and without guidance. The area of each bubble indicates the FLOPs for a single forward pass during training. Our method is more compute-efficient with competitive performance.

and implementation (Cory Stephenson, 2023), it is still inaccessible to most researchers and practitioners. Thus, improving the training efficiency of diffusion models is still an open question.

In the representation learning community, masked training is widely used to improve training efficiency in domains such as natural language processing (Devlin et al., 2018), computer vision (He et al., 2022), and vision-language understanding (Li et al., 2022b). Masked training significantly reduces the overall training time and memory, especially in vision applications (He et al., 2022; Li et al., 2022b). In addition, masked training is an effective self-supervised learning technique that does not sacrifice the representation learning quality, as high redundancy in the visual appearance allows the model to learn from unmasked patches. Here, masked training relies heavily on the transformer architecture (Vaswani et al., 2017) since they operate on patches, and it is natural to mask a subset of them.

Masked training, however, cannot be directly applied to diffusion models since current diffusion models mostly use U-Net (Ronneberger et al., 2015) as the standard network backbone, with some modifications (Ho et al., 2020; Song et al., 2021b; Dhariwal & Nichol, 2021). The convolutions in U-Net operate on regular dense grids, making it challenging to incorporate masked tokens and train with a random subset of the input patches. Thus, applying masked training techniques to U-Net models is not straightforward.

Recently, a few works (Peebles & Xie, 2022; Bao et al., 2023) show that replacing the U-Net architecture with vision transformers (ViT) (Dosovitskiy et al., 2020) as the backbone of diffusion models can achieve similar or even better performance across standard image generation benchmarks. In particular, Diffusion Transformer (DiT) (Peebles & Xie, 2022) inherits the best practices of ViTs without relying on the U-Net

inductive bias, demonstrating the scalability of the DiT backbone for diffusion models. Similar to standard ViTs, training DiTs is computationally expensive, requiring millions of iterations. However, this opens up new opportunities for efficient training of large transformer-based diffusion models.

**In this work**, we leverage the transformer structure of DiTs to enable masked modeling for significantly faster and cheaper training. Our main hypothesis is that images contain significant redundancy in the pixel space. Thus, it is possible to train diffusion models by minimizing the denoising score matching (DSM) loss on only a subset of pixels. Masked training has two main advantages: 1) It enables us to pass only a small subset of image patches to the network, which largely reduces the computational cost per iteration. 2) It allows us to augment the training data with multiple views, which can improve the training performance, particularly in limited-data settings.

Driven by this hypothesis, we propose an efficient approach to train large transformer-based diffusion models, termed Masked Diffusion Transformer (MaskDiT). We first adopt an asymmetric encoder-decoder architecture, where the transformer encoder only operates on the unmasked patches, and a lightweight transformer decoder operates on the complete set of patches. We then design a new training objective, consisting of two parts: 1) predicting the score of unmasked patches via the DSM loss, and 2) reconstructing the masked patches of the input via the mean square error (MSE) loss. This provides a global knowledge of the complete image that prevents the diffusion model from overfitting to the unmasked patches. For simplicity, we apply the same masking ratio (*e.g.*, 50%) across all the diffusion timesteps during training. After training, we finetune the model without masking to close the distribution gap between training and inference and this incurs only a minimal amount of overhead (less than 6% of the total training cost).

By randomly removing 50% patches of training images, our method reduces the per-iteration training cost by $2\times$. With our proposed new architecture and new training objective, MaskDiT can also achieve similar performance to the DiT counterpart with fewer training steps, leading to much smaller overall training costs. In the class-conditional ImageNet-$256\times256$ generation benchmark, MaskDiT achieves an FID of 5.69 without guidance, which outperforms previous (non-cascaded) diffusion models, and it achieves an FID of 2.28 with guidance, which is comparable to the state-of-the-art models. To achieve the result of FID 2.28, the total training cost of MaskDiT is 273 hours on $8\times$ A100 GPUs, which is only 31% of the DiT's training time. For class-conditional ImageNet-$512\times512$, MaskDiT achieves an FID of 10.79 without guidance and 2.50 with guidance, outperforming DiT-XL/2 and prior diffusion models. The total training cost is 209 A100 GPU days, which is about 29% of that of DiT-XL/2, validating the scalability and effectiveness of MaskDiT. All these results show that our method provides a significantly better training efficiency in wall-clock time.

Our main contributions are summarized as follows:

- We propose a new approach for fast training of diffusion models with masked transformers by randomly masking out a high proportion (50%) of input patches.

- For masked training, we introduce an asymmetric encoder-decoder architecture and a new training objective that predicts the score of unmasked patches and reconstructs masked patches.

- We show our method has faster training speed and less memory consumption while achieving comparable generation performance, implying its scalability to large diffusion models.

## 2 Related work

**Backbone architectures of diffusion models**  The success of diffusion models in image generation does not come true without more advanced backbone architectures. Since DDPM (Ho et al., 2020), the convolutional U-Net architecture (Ronneberger et al., 2015) has become the standard choice of diffusion backbone (Rombach et al., 2022; Saharia et al., 2022b). Built on the U-Net proposed in (Ho et al., 2020), several improvements have also been proposed (*e.g.*, (Song et al., 2021b; Dhariwal & Nichol, 2021)) to achieve better generation performance, including increasing depth versus width, applying residual blocks from BigGAN (Brock et al., 2019), using more attention layers and attention heads, rescaling residual connections with $\frac{1}{\sqrt{2}}$, etc. More recently, several works have proposed transformer-based architectures for

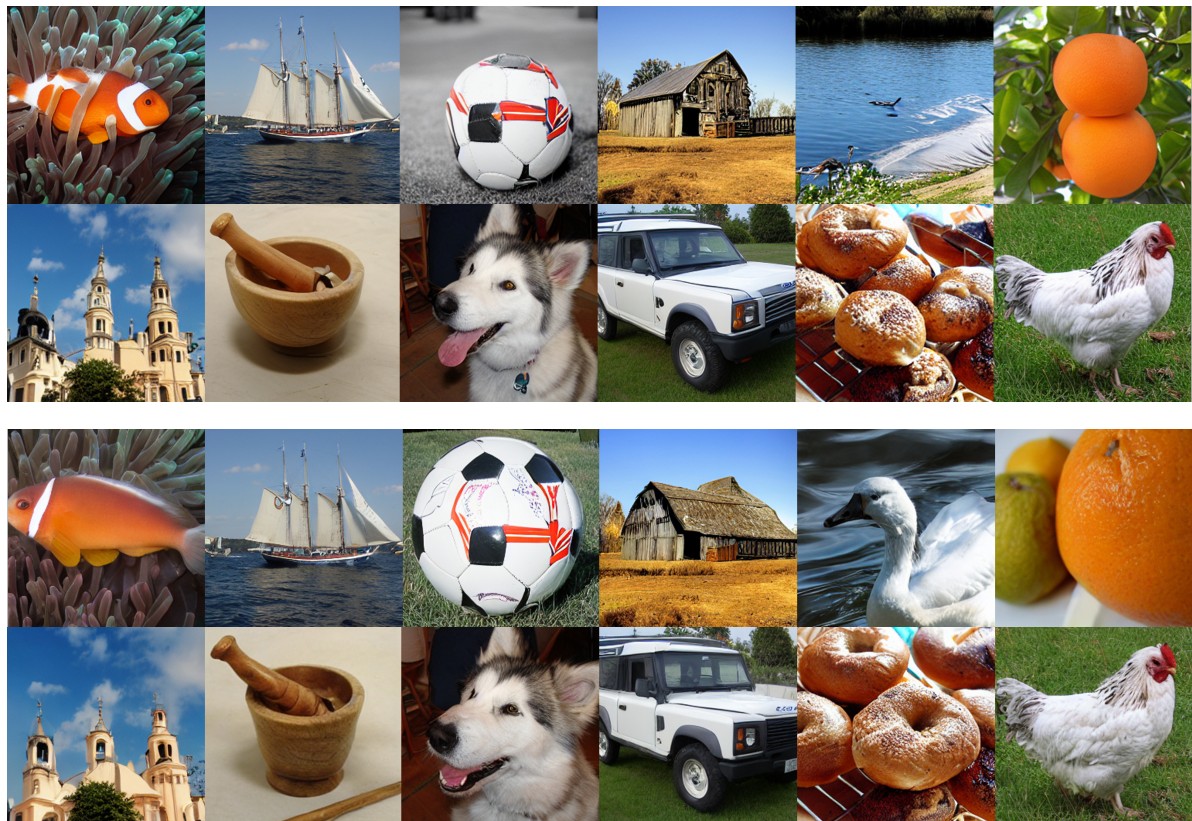

Figure 3: Generated samples from MaskDiT. Upper panel: without CFG. Lower panel: with CFG (scale=1.5). Both panels use the same batch of input latent noises and class labels.

diffusion models, such as GenViT (Yang et al., 2022), U-ViT (Bao et al., 2023), RIN (Jabri et al., 2022) and DiT (Peebles & Xie, 2022), largely due to the wide expressivity and flexibility of transformers (Vaswani et al., 2017; Dosovitskiy et al., 2020). Among them, DiT is a pure transformer architecture without the U-Net inductive bias and it obtains a better class-conditional generation result than the U-Net counterparts with better scalability (Peebles & Xie, 2022). Rather than developing a better transformer architecture for diffusion models, our main goal is to improve the training efficiency of diffusion models by applying the masked training to the diffusion transformers (*e.g.*, DiT).

**Efficiency in diffusion models**   Despite the superior performance of diffusion models, they suffer from high training and inference costs, especially on large-scale, high-resolution image datasets. Most works have focused on improving the sampling efficiency of diffusion models, where some works (Song et al., 2021a; Lu et al., 2022; Karras et al., 2022) improve the sampling strategy with advanced numerical solvers while others (Salimans & Ho, 2022; Meng et al., 2022; Zheng et al., 2022; Song et al., 2023) train a surrogate network by distillation. However, they are not directly applicable to reducing the training costs of diffusion models. For better training efficiency, Vahdat et al. (2021); Rombach et al. (2022) have proposed to train a latent-space diffusion model by first mapping the high-resolution images to the low-dimensional latent space. (Ho et al., 2022) developed a cascaded diffusion model, consisting of a base diffusion model on low-resolution images and several lightweight super-resolution diffusion models. More similarly, prior work (Wang et al., 2023) proposed a patch-wise training framework where the score matching is performed at the patch level. Different from our work that uses a subset of small patches as the transformer input, (Wang et al., 2023) uses a single large patch to replace the original image and only evaluates their method on the U-Net architecture.

**Masked training with transformers**   As transformers become the dominant architectures in natural language processing (Vaswani et al., 2017) and computer vision (Dosovitskiy et al., 2020), masked training

has also been broadly applied to representation learning (Devlin et al., 2018; He et al., 2022; Li et al., 2022b) and generative modeling (Radford et al., 2018; Chang et al., 2022; 2023) in these two domains. One of its notable applications is masked language modeling proposed by BERT (Devlin et al., 2018). In computer vision, a line of works has adopted the idea of masked language modeling to predict masked discrete tokens by first tokenizing the images into discrete values via VQ-VAE (Van Den Oord et al., 2017), including BEiT (Bao et al., 2022) for self-supervised learning, MaskGiT (Chang et al., 2022) and MUSE (Chang et al., 2023) for image synthesis, and MAGE (Li et al., 2022a) for unifying representation learning and generative modeling. On the other hand, MAE (He et al., 2022) applied masked image modeling to directly predict the continuous tokens (without a discrete tokenizer). This design also takes advantage of masking to reduce training time and memory and has been applied to various other tasks (Feichtenhofer et al., 2022; Geng et al., 2022). Our work is inspired by MAE, and it shows that masking can significantly improve the training efficiency of diffusion models when applying it to an asymmetric encoder-decoder architecture.

**Masked training of diffusion models** More recently, it starts to attract more attention to combine masked training and diffusion models. Wei et al. (2023) formulated diffusion models as masked autoencoders (DiffMAE) by denoising masked patches corrupted with Gaussian noises at different levels. But DiffMAE mainly focuses on the discriminative tasks, besides being able to solve the inpainting task. Gao et al. (2023) recently proposed MDT that adds the masked training objective to the original DiT loss for better generative performance. Our method has several differences with MDT: First, MDT still keeps the original DSM loss for training which takes full patches. In contrast, MaskDiT applies DSM only to unmasked tokens and proposes a reconstruction loss on masked patches, which leads to more efficient training. Second, MDT relies on extra modules like side-interpolater and relative positional bias while MaskDiT only needs minimal changes to the original DiT architecture. Last but not least, our main goal is fast training of diffusion models with little loss of performance while MDT aims at better generative performance with additional network modules and more computational cost per training iteration.

## 3 Method

In this section, we first introduce the preliminaries of diffusion models that we have adopted during training and inference, and then focus on the key designs of our proposed MaskDiT.

### 3.1 Preliminaries

**Diffusion models** We follow Song et al. (2021b), which introduces the forward diffusion and backward denoising processes of diffusion models in a continuous time setting using differential equations. In the forward process, diffusion models diffuse the real data $\boldsymbol{x}_0 \sim p_{\text{data}}(\boldsymbol{x}_0)$ towards a noise distribution $\boldsymbol{x}_T \sim \mathcal{N}(0, \sigma_{\max}^2 \mathbf{I})$ through the following stochastic differential equation (SDE):

$$d\boldsymbol{x} = \boldsymbol{f}(\boldsymbol{x}, t)dt + g(t)d\boldsymbol{w}, \tag{1}$$

where $\boldsymbol{f}$ is the (vector-valued) drift coefficient, $g$ is the diffusion coefficient, $\boldsymbol{w}$ is a standard Wiener process, and the time $t$ flows from 0 to $T$. In the reverse process, sample generation can be done by the following SDE:

$$d\boldsymbol{x} = [\boldsymbol{f}(\boldsymbol{x}, t) - g(t)^2 \nabla_{\boldsymbol{x}} \log p_t(\boldsymbol{x})]dt + g(t)d\bar{\boldsymbol{w}} \tag{2}$$

where $\bar{\mathbf{w}}$ is a standard reverse-time Wiener process and $dt$ is an infinitesimal negative timestep. The reverse SDE (2) can be converted to a probability flow ordinary differential equation (ODE) (Song et al., 2021b):

$$d\boldsymbol{x} = [\boldsymbol{f}(\boldsymbol{x}, t) - \frac{1}{2}g(t)^2 \nabla_{\boldsymbol{x}} \log p_t(\boldsymbol{x})]dt \tag{3}$$

which has the same marginals $p_t(\boldsymbol{x})$ as the forward SDE (1) at all timesteps $t$. We closely follow the EDM formulation (Karras et al., 2022) by setting $\boldsymbol{f}(\boldsymbol{x}, t) := \mathbf{0}$ and $g(t) := \sqrt{2t}$. Specifically, the forward SDE reduces to $\boldsymbol{x} = \boldsymbol{x}_0 + \boldsymbol{n}$ where $\boldsymbol{n} \sim \mathcal{N}(\mathbf{0}, t^2\mathbf{I})$, and the probability flow ODE becomes

$$d\boldsymbol{x} = -t\nabla_{\boldsymbol{x}} \log p_t(\boldsymbol{x})dt \tag{4}$$

To learn the score function $s(x, t) := \nabla_x \log p_t(x)$, EDM parameterizes a denoising function $D_\theta(x, t)$ to minimize the denoising score matching loss:

$$\mathbb{E}_{x_0 \sim p_{\text{data}}} \mathbb{E}_{n \sim \mathcal{N}(0, t^2 I)} \| D_\theta(x_0 + n, t) - x_0 \|^2 \tag{5}$$

and then the estimate score $\hat{s}(x, t) = (D_\theta(x, t) - x)/t^2$.

**Classifier-free guidance** For class-conditional generation, classifier-free guidance (CFG) (Ho & Salimans, 2022) is a widely used sampling method to improve the generation quality of diffusion models. In the EDM formulation, denote by $D_\theta(x, t, c)$ the class-conditional denoising function, CFG defines a modified denoising function: $\hat{D}_\theta(x, t, c) = D_\theta(x, t) + w(D_\theta(x, t, c) - D_\theta(x, t))$, where $w \geq 1$ is the guidance scale. To get the unconditional model for the CFG sampling, we can simply use a null token $\varnothing$ (*e.g.*, an all-zero vector) to replace the class label $c$, *i.e.*, $D_\theta(x, t) := D_\theta(x, t, \varnothing)$. During training, we randomly set $c$ to the null token $\varnothing$ with some fixed probability $p_{\text{uncond}}$.

### 3.2 Key designs

**Image masking** Given a clean image $x_0$ and the diffusion timestep $t$, we first get the diffused image $x_t$ by adding the Gaussian noise $n$. We then divide (or "patchify") $x_t$ into a grid of $N$ non-overlapping patches, each with a patch size of $p \times p$. For an image of resolution $H \times W$, we have $N = (HW)/p^2$. With a fixed masking ratio $r$, we randomly remove $\lfloor rN \rfloor$ patches and only pass the remaining $N - \lfloor rN \rfloor$ unmasked patches to the diffusion model. For simplicity, we keep the same masking ratio $r$ for all diffusion timesteps. A high masking ratio largely improves the computation efficiency but may also reduce the learning signal of the score estimation. Given the large redundancy of $x_t$, the learning signal with masking may be compensated by the model's ability to extrapolate masked patches from neighboring patches. Thus, there may exist an optimal spot where we achieve both good performance and high training efficiency.

**Asymmetric encoder-decoder backbone** Our diffusion backbone is based on DiT, a standard ViT-based architecture for diffusion models, with a few modifications. Similar to MAE (He et al., 2022), we apply an asymmetric encoder-decoder architecture: 1) the encoder has the same architecture as the original DiT except without the final linear projection layer, and it only operates on the unmasked patches; 2) the decoder is another DiT architecture adapted from the lightweight MAE decoder, and it takes the full tokens as the input. Similar to DiT, our encoder embeds patches by a linear projection with standard ViT frequency-based positional embeddings added to all input tokens. Then masked tokens are removed before being passed to the remaining encoder layers. The decoder takes both the encoded unmasked tokens in addition to new `mask` tokens as input, where each `mask` token is a shared, learnable vector. We then add the same positional embeddings to all tokens before passing them to the decoder. Due to the asymmetric design (*e.g.*, the MAE decoder with <9% parameters of DiT-XL/2), masking can dramatically reduce the computational cost per iteration.

**Training objective** Unlike the general training of diffusion models, we do not perform denoising score matching on full tokens. This is because it is challenging to predict the score of the masked patches by solely relying on the visible unmasked patches. Instead, we decompose the training objective into two subtasks: 1) the score estimation task on unmasked tokens and 2) the auxiliary reconstruction task on masked tokens. Formally, denote the binary masking label by $m \in \{0, 1\}^N$. We uniformly sample $\lfloor rN \rfloor$ patches without replacement and mask them out. The denoising score matching loss on unmasked tokens reads:

$$\mathcal{L}_{\text{DSM}} = \mathbb{E}_{x_0 \sim p_{\text{data}}} \mathbb{E}_{n \sim \mathcal{N}(0, t^2 I)} \mathbb{E}_m \| (D_\theta((x_0 + n) \odot (1 - m), t) - x_0) \odot (1 - m) \|^2 \tag{6}$$

where $\odot$ denotes the element-wise multiplication along the token length dimension of the "patchified" image, and $(x_0 + n) \odot (1 - m)$ denotes the model $D_\theta$ only takes the unmasked tokens as input. Similar to MAE, the reconstruction task is performed by the MSE loss on the masked tokens:

$$\mathcal{L}_{\text{MAE}} = \mathbb{E}_{x_0 \sim p_{\text{data}}, n \sim \mathcal{N}(0, t^2 I)} \mathbb{E}_m \| (D_\theta((x_0 + n) \odot (1 - m), t) - (x_0 + n)) \odot m \|^2 \tag{7}$$

where the goal is to reconstruct the input (*i.e.*, the diffused image $\boldsymbol{x}_0 + \boldsymbol{n}$) by predicting the pixel values for each masked patch. We hypothesize that adding this MAE reconstruction loss can promote the masked transformer to have a global understanding of the full (diffused) image and thus prevent the denoising score matching loss in (6) from overfitting to a local subset of visible patches.

Finally, our training objective is given by

$$\mathcal{L} = \mathcal{L}_{\text{DSM}} + \lambda \mathcal{L}_{\text{MAE}} \tag{8}$$

where the hyperparameter $\lambda$ controls the balance between the score prediction and MAE reconstruction losses, and it cannot be too large to drive the training away from the standard DSM update.

**Unmasking tuning** Although the model is trained on masked images, it can be used to generate full images with high quality in the standard class-conditional setting. However, we observe that the masked training always leads to a worse performance of the CFG sampling than the original DiT training (without masking). This is because classifier-free guidance relies on both conditional and unconditional score. Without label information, learning unconditional score from only 50% tokens is much harder than that of conditional score. Therefore, the unconditional model (by setting the class label $c$ to the null token $\varnothing$) is not as good as the conditional model. To further close the gap between the training and inference, after the masked training, we finetune the model using a smaller batch size and learning rate at minimal extra cost. We also explore two different masking ratio schedules during unmasking tuning: 1) the *zero-ratio* schedule, where the masking ratio $r = 0$ throughout all the tuning steps, and 2) the *cosine-ratio* schedule, where the masking ratio $r = 0.5 * \cos^4(\pi/2 * i/n_{\text{tot}})$ with $i$ and $n_{\text{tot}}$ being the current steps and the total number of tuning steps, respectively. This unmasking tuning strategy trades a slightly higher training cost for the better performance of CFG sampling.

## 4 Experiments

### 4.1 Experimental setup

**Model settings** Following DiT (Peebles & Xie, 2022), we consider the latent diffusion model (LDM) framework, whose training consists of two stages: 1) training an autoencoder to map images to the latent space, and 2) training a diffusion model on the latent variables. For the autoencoder, we use the pre-trained VAE model from Stable Diffusion (Rombach et al., 2022), with a downsampling factor of 8. Thus, an image of resolution $256 \times 256 \times 3$ results in a $32 \times 32 \times 4$ latent variable while an image of resolution $512 \times 512 \times 3$ results in a $64 \times 64 \times 4$ latent variable. For the diffusion model, we use DiT-XL/2 as our encoder architecture, where "XL" denotes the largest model size used by DiT and "2" denotes the patch size of 2 for all input patches. Our decoder has the same architecture with MAE (He et al., 2022), except that we add the adaptive layer norm blocks for conditioning on the time and class embeddings (Peebles & Xie, 2022).

**Training details** Most training details are kept the same with the DiT work: AdamW (Loshchilov & Hutter, 2017) with a constant learning rate of 1e-4, no weight decay, and an exponential moving average (EMA) of model weights over training with a decay of 0.9999. Also, we use the same initialization strategies with DiT. By default, we use a masking ratio of 50%, an MAE coefficient $\lambda = 0.1$, a probability of dropping class labels $p_{\text{uncond}} = 0.1$, and a batch size of 1024. Because masking greatly reduces memory usage and FLOPs, we find it more efficient to use a larger batch size during training. Unlike DiT which uses horizontal flips, we do not use any data augmentation, as our early experiments showed it does not bring better performance. For the unmasked tuning, we change the learning rate to 5e-5 and use full precision for better training stability. Unless otherwise noted, experiments on ImageNet $256 \times 256$ are conducted on $8\times$ A100 GPUs, each with 80GB memory, whereas for ImageNet $512 \times 512$, we use $32\times$ A100 GPUs.

**Evaluation metrics** Following DiT (Peebles & Xie, 2022), we mainly use Fréchet Inception Distance (FID) (Heusel et al., 2017) to guide the design choices by measuring the generation diversity and quality of our method, as it is the most commonly used metric for generative models. To ensure a fair comparison with previous methods, we also use the ADM's TensorFlow evaluation suite (Dhariwal & Nichol, 2021) to compute FID-50K with the same reference statistics. Similarly, we also use Inception Score (IS) (Salimans et al., 2016),

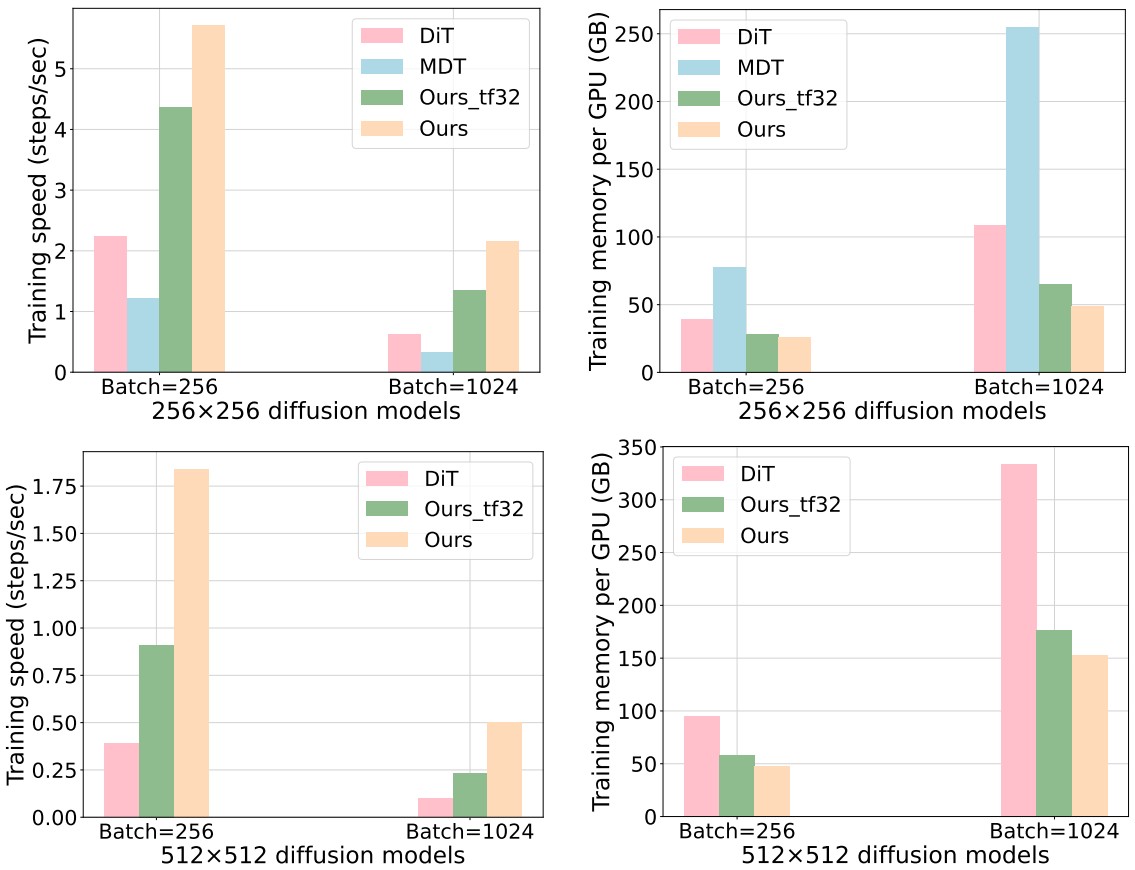

Figure 4: Training speed (in steps/sec) and memory-per-GPU (in GB) for DiT, MDT and MaskDiT (Ours), measured on a single node with 8×A100 GPUs, each with 80 GB memory. (Top): Results on ImageNet-256×256. (Bottom): Results on ImageNet 512×512. Here we use the latest official implementation of DiT and MDT where DiT applies the TensorFloat32 (TF32) precision and MDT applies the Float32 precision. Our MaskDiT, by default, applies Mixed Precision. We also add the TF32 precision of MaskDiT (termed "Ours_ft32") for reference. If the GPU memory requirements surpass the GPU hardware of our hardware, we enable gradient accumulation and estimate the GPU memory usages accounting for typical overheads in distributed training communication. MDT paper has no results on 512×512 so we exclude it from the comparison on 512×512.

sFID (Nash et al., 2021) and Precision/Recall (Kynkäänniemi et al., 2019) as secondary metrics to compare with previous methods. Note that for FID and sFID, the lower is better while for IS and Precision/Recall, the higher is better. Without stating explicitly, the FID values we report in experiments refer to the cases without classifier-free guidance.

## 4.2 Training efficiency

We compare the training efficiency of MaskDiT, DiT-XL/2 (Peebles & Xie, 2022), and MDT-XL/2 (Gao et al., 2023) on 8× A100 GPUs, from three perspectives: GLOPs, training speed and memory consumption, and wall-time training convergence. All three models have a similar model size.

First, in Figure 2, the GFLOPs of MaskDiT, defined as FLOPs for a single forward pass during training, are much fewer than DiT and MDT. Specifically, our GFLOPs are only 54.0% of DiT and 31.7% of MDT. As a reference, LDM-8 has a similar number of GFLOPs with MaskDiT, but its FID is much worse than MaskDiT. Second, in Figure 4, the training speed of MaskDiT is much larger than the two previous methods while our memory-per-GPU cost is much lower, no matter whether Mixed Precision is applied. Also, the

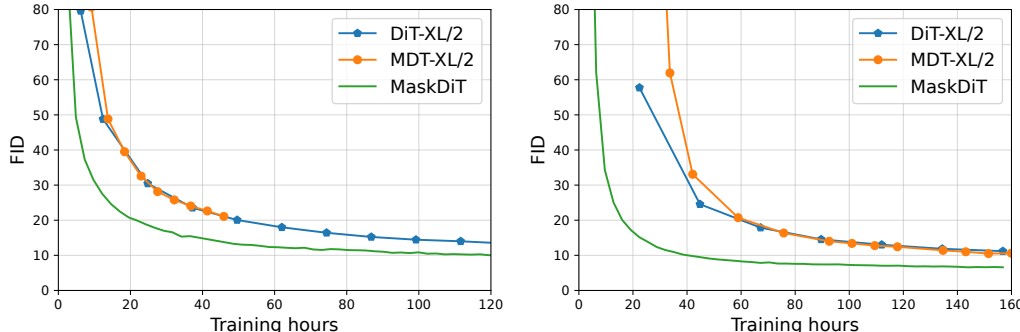

Figure 5: FID vs. training hours on ImageNet 256×256. Left: batch size 256. Right: batch size 1024. For DiT and MDT, we use their official implementations with default settings. For batch size 256, we only report the FID of MDT before 220k steps because its gradient explodes after that.

Table 1: Comparison of MaskDiT with various state-of-the-art generative models on class-conditional ImageNet 256×256, where -G means the classifier-free guidance.

| Method | FID↓ | sFID↓ | IS↑ | Prec.↑ | Rec.↑ |
|---|---|---|---|---|---|
| BigGAN-deep (Brock et al., 2018) | 6.95 | 7.36 | 171.40 | 0.87 | 0.28 |
| StyleGAN-XL (Sauer et al., 2022) | 2.30 | 4.02 | 265.12 | 0.78 | 0.53 |
| MaskGIT (Chang et al., 2022) | 6.18 | - | 182.10 | 0.80 | 0.51 |
| CDM (Ho et al., 2022) | 4.88 | - | 158.71 | - | - |
| ADM (Dhariwal & Nichol, 2021) | 10.94 | 6.02 | 100.98 | 0.69 | 0.63 |
| ADM-U | 7.49 | **5.13** | 127.49 | 0.72 | 0.63 |
| LDM-8 (Rombach et al., 2022) | 15.51 | - | 79.03 | 0.65 | 0.63 |
| LDM-4 | 10.56 | - | **209.52** | **0.84** | 0.35 |
| U-ViT-H/2 (Bao et al., 2023) | 6.58 | - | - | - | - |
| DiT-XL/2 (Peebles & Xie, 2022) | 9.62 | 6.85 | 121.50 | 0.67 | **0.67** |
| MDT-XL/2 (Gao et al., 2023) | 6.23 | 5.23 | 143.02 | 0.71 | 0.65 |
| **MaskDiT** | **5.69** | 10.34 | 177.99 | 0.74 | 0.60 |
| ADM-G (Dhariwal & Nichol, 2021) | 4.59 | 5.25 | 186.70 | 0.82 | 0.52 |
| ADM-G, ADM-U | 3.94 | 6.14 | 215.84 | 0.83 | 0.53 |
| LDM-8-G (Rombach et al., 2022) | 7.76 | - | 103.49 | 0.71 | **0.62** |
| LDM-4-G | 3.60 | - | 247.67 | **0.87** | 0.48 |
| U-ViT-H/2-G (Bao et al., 2023) | 2.29 | 5.68 | 263.88 | 0.82 | 0.57 |
| DiT-XL/2-G (Peebles & Xie, 2022) | 2.27 | 4.60 | 278.24 | 0.83 | 0.57 |
| MDT-XL/2-G (Gao et al., 2023) | **1.79** | **4.57** | **283.01** | 0.81 | 0.61 |
| **MaskDiT-G** | 2.28 | 5.67 | 276.56 | 0.80 | 0.61 |

improvement becomes higher as we use a larger batch size. For resolution 256×256 with a batch size of 1024, our training speed is 3.5× of DiT and 6.5× of MDT, while our memory cost is only 45.0% of DiT's and 19.2% of MDT's. For resolution 512×512 with a batch size of 1024, our training speed is 4.6× of DiT's, and our GPU memory cost remains as low as 45.7% of DiT's. Finally, in Figure 5, MaskDiT also consistently yields better wall-time training convergence than DiT and MDT with different batch sizes. Similarly, increasing the batch size further improves the efficiency of MaskDiT, but it does not benefit DiT-XL/2 and MDT-XL/2. For instance on ImageNet 256×256, when using a batch size of 1024, MaskDiT can achieve an FID of 10 within 40 hours, while the other two methods take more than 160 hours, implying a 4× speedup.

Table 2: Comparison of MaskDiT with various state-of-the-art generative models on class-conditional ImageNet 512×512, where -G means the classifier-free guidance.

| Method | FID↓ | sFID↓ | IS↑ | Prec.↑ | Rec.↑ |
|---|---|---|---|---|---|
| BigGAN-deep (Brock et al., 2018) | 8.43 | 8.13 | 177.90 | 0.88 | 0.29 |
| StyleGAN-XL (Sauer et al., 2022) | 2.41 | 4.06 | 267.75 | 0.77 | 0.52 |
| ADM (Dhariwal & Nichol, 2021) | 23.24 | 10.19 | 58.06 | 0.73 | 0.60 |
| ADM-U | **9.96** | **5.62** | 121.78 | **0.75** | **0.64** |
| DiT-XL/2 (Peebles & Xie, 2022) | 12.03 | 7.12 | 105.25 | **0.75** | **0.64** |
| **MaskDiT** | 10.79 | 13.41 | **145.08** | 0.74 | 0.56 |
| ADM-G (Dhariwal & Nichol, 2021) | 7.72 | 6.57 | 172.71 | **0.87** | 0.42 |
| ADM-G, ADM-U | 3.85 | 5.86 | 221.72 | 0.84 | 0.53 |
| DiT-XL/2-G (Peebles & Xie, 2022) | 3.04 | **5.02** | 240.82 | 0.84 | 0.54 |
| **MaskDiT-G** | **2.50** | 5.10 | **256.27** | 0.83 | **0.56** |

## 4.3 Comparison with state-of-the-art

We compare against state-of-the-art class-conditional generative models in Table 1 and 2, where our 256×256 results are obtained after 2M training steps and 512×512 results are obtained after 1M training steps.

**ImageNet-256×256.** Without CFG, we further perform the unmasking tuning with the *cosine-ratio* schedule for 37.5k steps. MaskDiT achieves a better FID than all the non-cascaded diffusion models, and it even outperforms the MDT-XL/2, which is much more computationally expensive, by decreasing FID from 6.23 to 5.69. Although the cascaded diffusion model (CDM) has a better FID than MaskDiT, it underperforms MaskDiT regarding IS (158.71 vs. 177.99).

When using CFG, we perform the unmasking tuning with the *zero-ratio* schedule for 75k steps. MaskDiT-G achieves a very similar FID (2.28) with DiT-XL/2-G (2.27), and they both share similar values of IS and Precision/Recall. Notably, MaskDiT achieves an FID of 2.28 with only 273 hours (257 hours for training and 16 hours for unmasking tuning) on 8× A100 GPUs while DiT-XL/2 (7M steps) takes 868 hours, implying 31% of the DiT-XL/2's training time. Compared with the MDT-XL/2-G, MaskDiT-G performs worse in terms of FID and IS but has comparable Precision/Recall scores.

**ImageNet-512×512.** As shown in Table 2, without CFG, MaskDiT achieves an FID of 10.79, outperforming DiT-XL/2 that has an FID of 12.03. With CFG, MaskDiT-G achieves an FID of 2.50, outperforming all previous methods, including ADM (FID: 3.85) and DiT (FID: 3.04). The total training cost including unmasked tuning is around 209 A100 GPU days, which is about 29% of the training cost of DiT (712 A100 GPU days). We provide 512×512 MaskDiT samples in Figure 11 and Figure 12.

We also observe that increasing the training budgets in the unmasking tuning stage can consistently improve the FID, but it also harms the training efficiency of our method. Given that our main goal is not to achieve new state-of-the-art and our generation quality is already good as demonstrated by Figure 3 and Appendix F we do not explore further in this direction.

## 4.4 Ablation studies

We conduct comprehensive ablation studies to investigate the impact of various design choices on our method. Unless otherwise specified, we use the default settings for training and inference, and report FID without guidance as the evaluation metric.

**Masking ratio and diffusion backbone** As shown in Table 3a, our asymmetric encoder-decoder architecture is always better than the original DiT architecture. When FID>100, we consider these two backbones to behave equally poorly. The performance gain from the asymmetric backbone is the most significant with

Table 3: Ablating our method. Unless stated otherwise, we use the following setting: our asymmetric encoder-decoder architecture, 50% masking ratio, the MAE coefficient $\lambda = 0$, and applying DSM on unmasked tokens only. By default, we report FID after training for 400k steps with a batch size of 1024.

(a) Impact of masking ratio and diffusion backbone

|  | mask 0% | mask 50% | mask 75% |
|---|---|---|---|
| DiT backbone | 14.70 | 24.58 | 107.55 |
| Our backbone | 13.71 | 12.31 | 121.16 |

(b) Impact of MAE reconstruction

|  | $\lambda = 0$ | 0.01 | 0.1 | 1.0 |
|---|---|---|---|---|
| 400k steps | 12.31 | 13.76 | 8.87 | 9.76 |
| 800k steps | 12.42 | 12.93 | 7.19 | 12.74 |

(c) Impact of DSM design

|  | $\lambda = 0$ | 0.1 |
|---|---|---|
| DSM on unmasked tokens | 12.31 | 8.87 |
| DSM on full tokens | 10.59 | 9.72 |

a proper masking ratio (*i.e.*, 50%). Without masking, the performance gain (FID: 14.70 → 13.71) mainly comes from the larger model size brought by the newly added decoder layers, confirming the observed expressivity of DiT-like architectures (Peebles & Xie, 2022). With a masking ratio of 50%, we see a large performance decrease (FID: 14.70 → 24.58) in the original DiT architecture but a performance gain (FID: 13.71 → 12.31) in the asymmetric encoder-decoder architecture. It implies that the interplay of the image masking and the asymmetric diffusion backbone contributes to the success of our method. Finally, the failure of masking 75% suggests that we cannot arbitrarily increase the masking ratio for higher efficiency.

**MAE reconstruction** As shown in Table 3b, adding the MAE reconstruction auxiliary task can largely benefit the generation quality of our method (*e.g.*, FID from 12.42 to 7.19 at 800k steps). Moreover, the MAE coefficient $\lambda = 0.1$ works the best, where we also observe the consistent improvement of FID over training iterations. A carefully tuned the MAE coefficient is crucial for our method: On one hand, if the MAE coefficient is too large (*i.e.,* $\lambda = 1.0$), FID first decreases (FID=9.76 at 400k steps) but then increases (FID=12.74 at 800k steps) during training. We hypothesize that the training in the late stage has been gradually dominated by the MAE reconstruction task, and the MAE training alone does not produce photorealistic images (He et al., 2022). On the other hand, if the MAE coefficient is too small (*i.e.,* $\lambda = 0$), the FID improvement seems to be saturated at the early training stage, as evidenced by FID=12.31 at 400k steps versus FID=12.42 at 800k steps. It confirms our intuition that without the MAE reconstruction task, the training easily overfits the local subset of unmasked tokens as it lacks a global understanding of the full image, making the gradient update less informative.

**DSM design** Since our default choice is to add the DSM loss on the unmasked tokens only, we compare its performance with adding the DSM loss on the full tokens. As shown in Table 3c, without the MAE reconstruction loss ($\lambda = 0$), adding the DSM loss on the unmasked tokens only performs worse than that on the full tokens. It can be explained by the fact that the score prediction on full tokens closes the gap between training and inference with our masking strategy. On the contrary, with the MAE reconstruction loss ($\lambda = 0.1$), adding the DSM loss on the unmasked tokens only performs better than that on the full tokens. It is because, in this setting, we jointly perform the score prediction and MAE reconstruction tasks on the masked, invisible tokens, which are contradictory to each other. Furthermore, we also observe that across these four settings in Table 3c, the best FID score is achieved by "DSM on unmasked tokens & $\lambda = 0.1$", which highlights the important role of the interplay between our DSM design and MAE reconstruction.

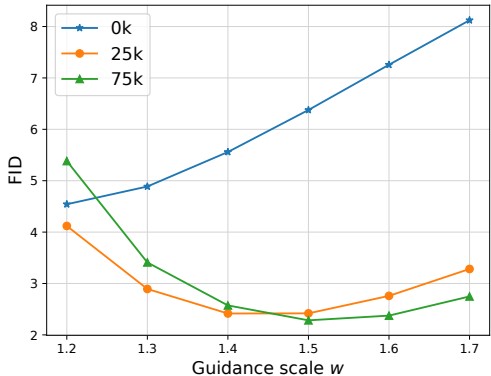

Figure 6: Impact of unmasking tuning on FID with guidance where we vary both the guidance scale $w$ and the number of tuning steps.

**Unmasking tuning**   To show the impact of unmasking tuning, we report FID with guidance of different scales in Figure 6, where we compare three tuning steps: 0k (no tuning), 25k, and 75k. We observe that without unmasking tuning, the best FID=4.54 is obtained by the guidance scale $w = 1.2$, and the score keeps increasing as we further increase $w$. As we perform the unmasking tuning, the best FID with guidance improves over iterations (*e.g.*, from FID=2.42 at 25k steps to FID=2.28 at 75k steps). Interestingly, the optimal guidance scale $w^*$ also shifts to a larger value over iterations (*e.g.*, from $w^* = 1.4$ at 25k steps to $w^* = 1.5$ at 75k steps). This is mainly because that masked training does not produce a good unconditional model, as evidenced by the observation that unconditional FID is 47.10 when training with a mask ratio of 50% for 500k steps and it becomes 32.19 without masking. However, unmasking tuning improves the unconditional model. Thus, we can use a large $w$ for a better FID as the number of tuning steps increases.

## 5   Broader impacts and limitations

Our work can reduce the training costs of diffusion models from the algorithmic perspective without sacrificing the generative performance. Our algorithmic innovation is orthogonal to the improved infrastructure and implementation (Cory Stephenson, 2023). A combination of these two improvements can further make the development of large diffusion models more accessible to the broad community. Although our work is directly tied to particular applications, MaskDiT shares with other image synthesis tools similar potential benefits and risks, such as generating Deepfakes for disinformation, which have been discussed extensively (*e.g.*, (Vaccari & Chadwick, 2020)).

One of our limitations is that we still need a few steps of unmasking tuning to match the state-of-the-art FID with guidance. It is directly related to another issue of masked training: it does not produce a good unconditional diffusion model. We leave it as future work to explore how to improve the unconditional image generation performance of MaskDiT.

## 6   Conclusions

In this work, we propose MaskDiT, an efficient approach to training diffusion models with masked transformers. We randomly mask out a large portion of the image patches, which significantly reduces the training overhead per iteration. To better accommodate masked training for diffusion models, we first introduce an asymmetric encoder-decoder diffusion backbone, where the DiT encoder only takes visible tokens and the lightweight DiT decoder takes full tokens after injecting masked tokens. We also incorporate an auxiliary reconstruction loss into the DSM objective, where we reconstruct the input values of masked tokens and predict the score of unmasked tokens. On the class-conditional ImageNet-256×256 and ImageNet-512×512 generation benchmarks, we demonstrated a better training efficiency for our proposed MaskDiT with a competitive performance with the state-of-the-art generative models.

## Acknowledgements

We thank Sangwoo Mo for sharing insights into the DiT training, and Zhiding Yu, Shiyi Lan and Boyi Li for helpful discussions. Anima Anandkumar is supported by Bren Chair professorship and Schmidt Sciences AI 2050 Senior fellow funding.

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

# Appendix

## A More experimental settings

**Diffusion configurations.** Different from DiT which is based on the ADM formulation (Dhariwal & Nichol, 2021), we use the EDM formulation (Karras et al., 2022) for training simplicity and inference efficiency. Specifically, we use the EDM preconditioning via a $\sigma$-dependent skip connection with the default hyperparameters (see (Karras et al., 2022) for more details). Thus, we do not need to learn ADM's parameterization of the noise covariance as in DiT. During inference, we use the default time schedule $t_{i<N} = (t_{\max}^{\frac{1}{\rho}} + \frac{i}{N-1}(t_{\min}^{\frac{1}{\rho}} - t_{\max}^{\frac{1}{\rho}}))^{\rho}$, where $N = 40$, $\rho = 7$, $t_{\max} = 80$ and $t_{\min} = 0.002$, and the second-order Heun's method as the ODE solver for sampling. Our early experiments with the original DiT architecture show that Heun's method reaches the same FID as 250 DDPM sampling steps used in DiT with a much lower number of function evaluations (79 vs. 250 evaluations), confirming the observations in (Karras et al., 2022).

**Training details.** We use the pre-trained VAE model ft-MSE with scaling factor 8 from the latent diffusion model (Rombach et al., 2022). To avoid repeatedly encoding the same images during training, we use the VAE encoder to encode the dataset into latent space and store them on the disk. All the experiments are running on this latent dataset, including the runs of DiT and MDT. For all the masked training of MaskDiT, we enable automatic mixed precision. During the unmasking tuning, we instead use TensorFloat32 for better numerical stability.

## B Benchmarking DiT and MDT

To make a fair comparison with DiT (Peebles & Xie, 2022) and MDT (Gao et al., 2023), we use their official implementations. Since DiT, MDT and our MaskDiT use the same VAE encoder, we replace the original dataset with the latent dataset to reduce the number of redundant encoding processes. We benchmark their highest-capacity models DiT-XL/2 and MDT-XL/2, with the default setting in their paper.

## C Compute used for the experiments

We run all the experiments on 8×A100 GPUs. The main experiment of MaskDiT takes 273 hours, of which 257 hours are spent on training and 16 hours on unmasking tuning. All the experiments in the ablation studies take around 1250 hours in total. The experiments on the training efficiency of MaskDiT, DiT-XL/2, and MDT-XL/2 take around 1000 hours in total. The preliminary or failed experiments not reported in the paper take around 800 hours in total.

# D  FID vs. training steps

Figure 7 plots the FID vs. training steps of the same experiments we report in Figure 5. MaskDiT, DiT-XL/2, and MDT-XL/2 are trained for the same amount of time. As shown in the figure, MaskDiT has a similar or even faster learning speed as DiT-XL/2 in terms of steps. The advantage of MaskDiT over DiT-XL/2 becomes more significant when the batch size is larger. Additionally, every training step of MaskDiT is around 45% cheaper than DiT-XL/2. Therefore MaskDiT is much more efficient than DiT-XL/2. MDT-XL/2 has a faster learning speed than DiT-XL/2 and MaskDiT, but its per-step cost is much higher. We do not report MDT's results after 220k steps in the setting of batch size 256 because we encountered a numerical issue while running MDT's code.

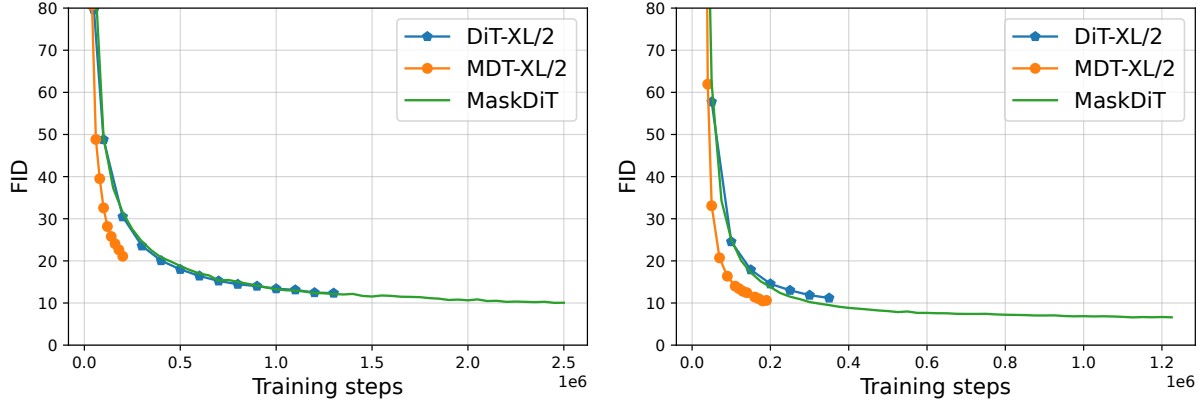

Figure 7: FID vs. training steps. Left: batch size 256. Right: batch size 1024. For batch size 256, we only reported the FID of MDT before 220k steps, because it encountered a gradient explosion issue afterwards when we ran the official MDT code with their default setting.

# E  Discussions on masking ratio schedule in unmasking tuning

In this paper, we explore two strategies for unmasking tuning. The first one is to disable masking during finetuning completely (*i.e.*, the *zero-ratio* schedule). This simple strategy effectively improves the FID with classifier-free guidance from 4.54 to 2.28. The second strategy adopts a cosine-ratio schedule where the mask ratio gradually decreases from 50% to 0%. More precisely, the masking ratio is calculated as $r = 0.5 * \cos^4(\pi/2 * i/n_{tot})$ with $i$ and $n_{tot}$ being the current steps and the total number of tuning steps, respectively. We find this strategy improves the FID without guidance from 5.95 to 5.69 within 37.5k steps. Clearly, different unmasking tuning strategies lead to different results. We do not fully explore this direction in this paper as it is not the focus of this paper. However, it would be an interesting future direction to investigate different tuning strategies for masked training of diffusion transformers.

# F  More generated samples

Figures 8, 9, and 10 show three diverse sets of images conditionally sampled from our MaskDiT with guidance 2.5 using 40 EDM (Karras et al., 2022) sampling steps.

"alp"

"black swan"

"daisy"

"Eskimo dog"

"hamster"

"orange"

"promontory"

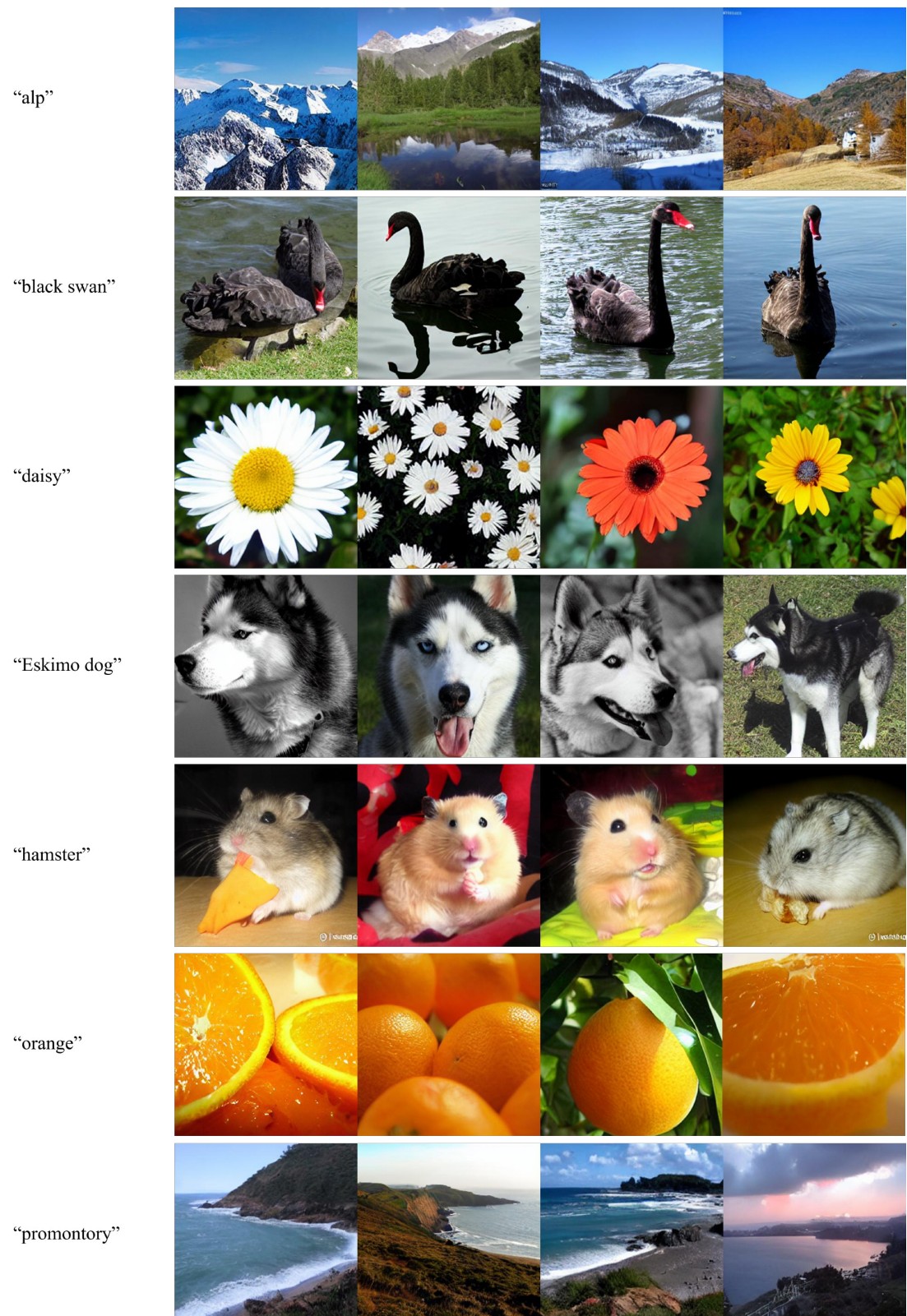

Figure 8: Class-conditional 256×256 image generation from MaskDiT with guidance 2.5 and 40 deterministic EDM (Karras et al., 2022) sampling steps. Every four images in a row are from the same class.

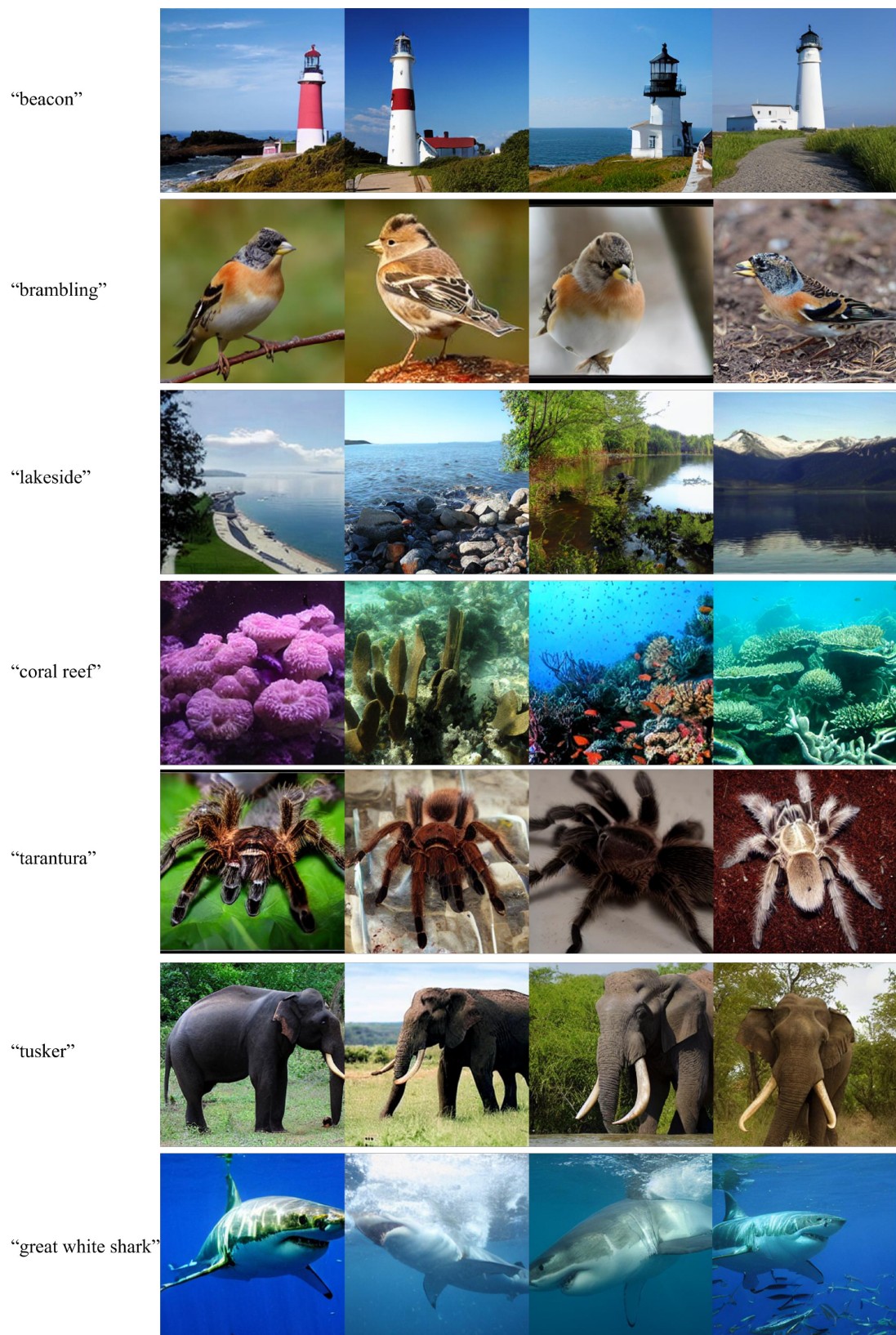

Figure 9: Class-conditional 256×256 image generation from MaskDiT with guidance 2.5 and 40 deterministic EDM (Karras et al., 2022) sampling steps. Every four images in a row are from the same class.

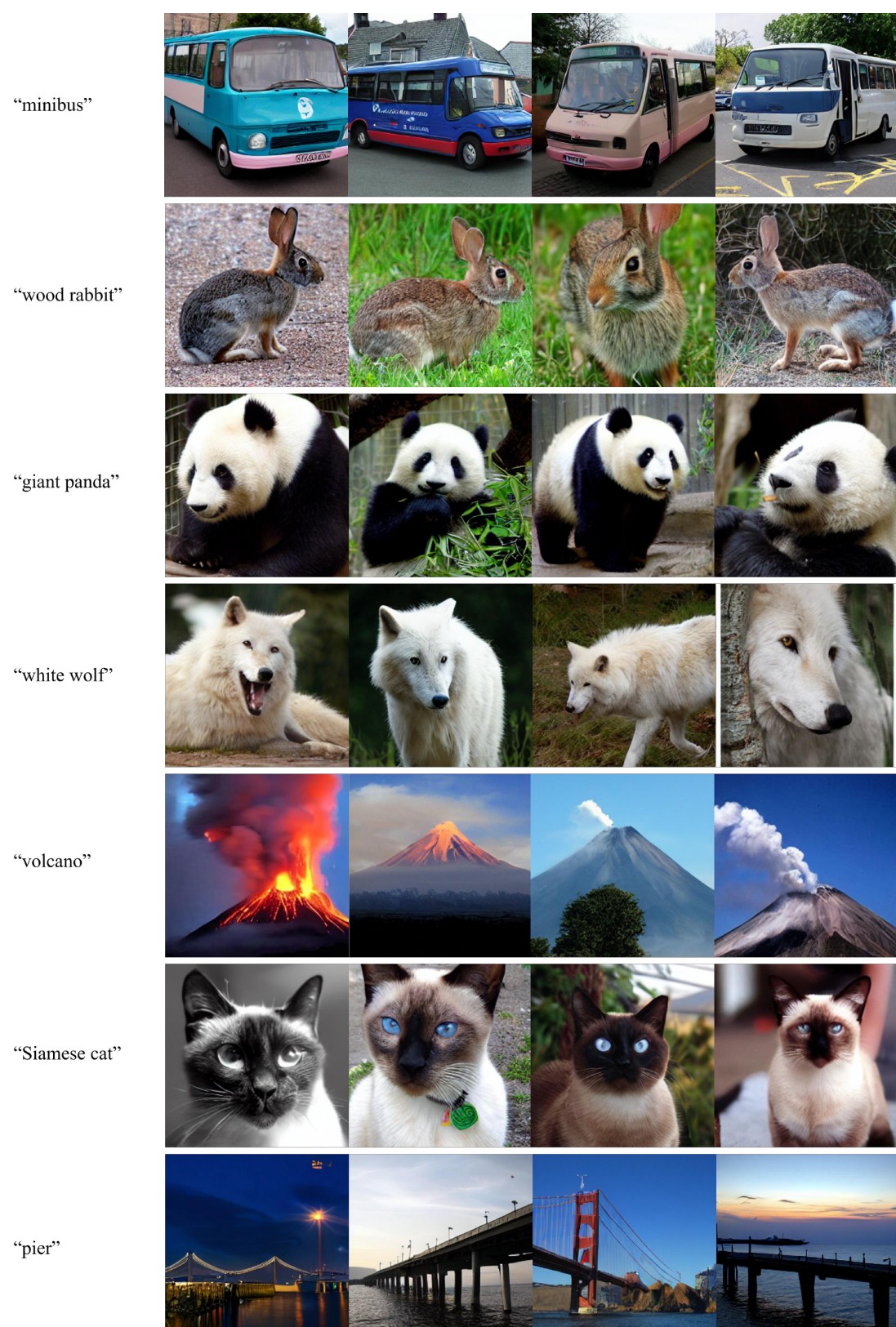

Figure 10: Class-conditional 256×256 image generation from MaskDiT with guidance 2.5 and 40 deterministic EDM (Karras et al., 2022) sampling steps. Every four images in a row are from the same class.

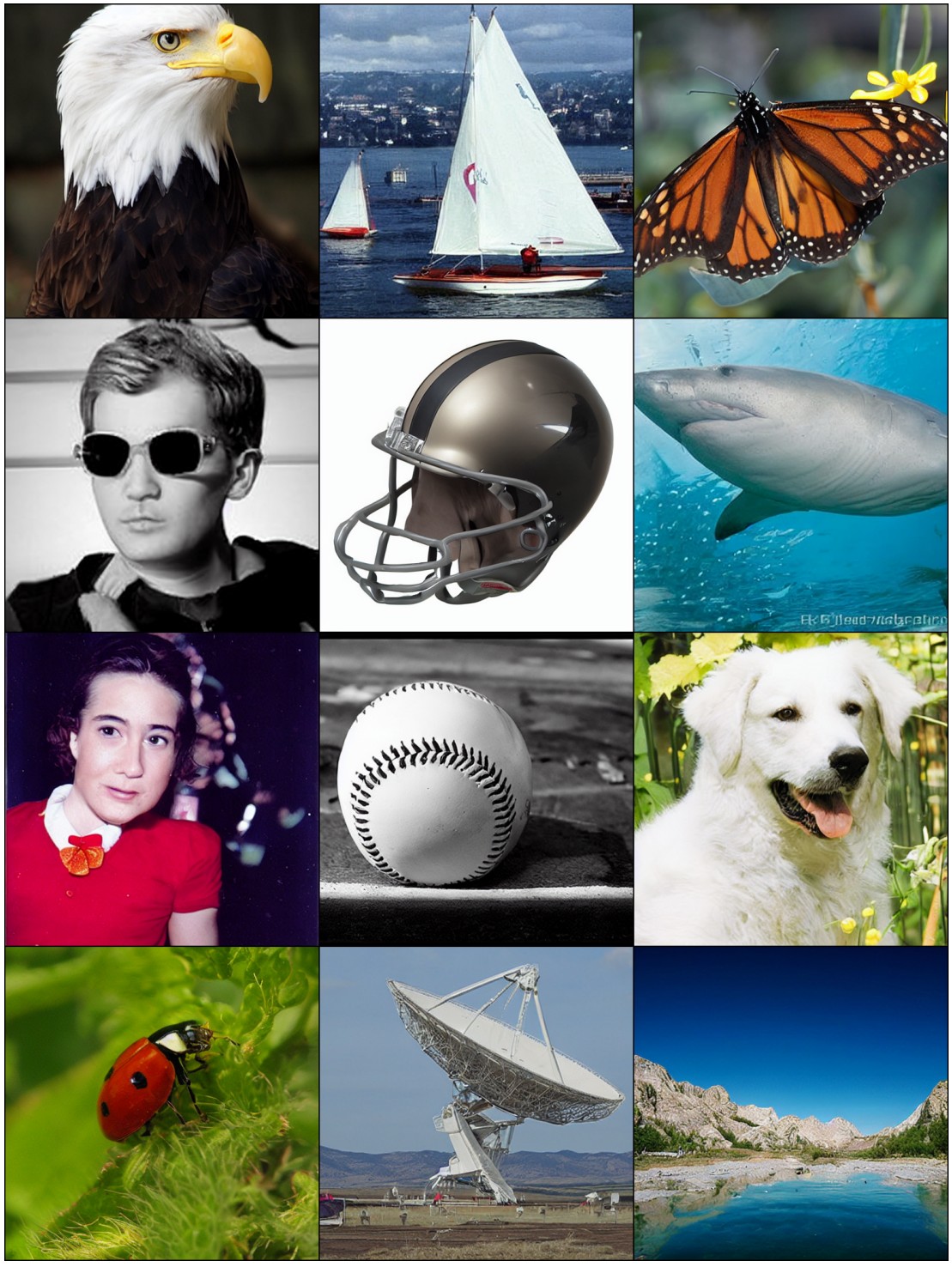

Figure 11: Class-conditional 512×512 image generation from MaskDiT with guidance 1.5 and 40 deterministic EDM (Karras et al., 2022) sampling steps.

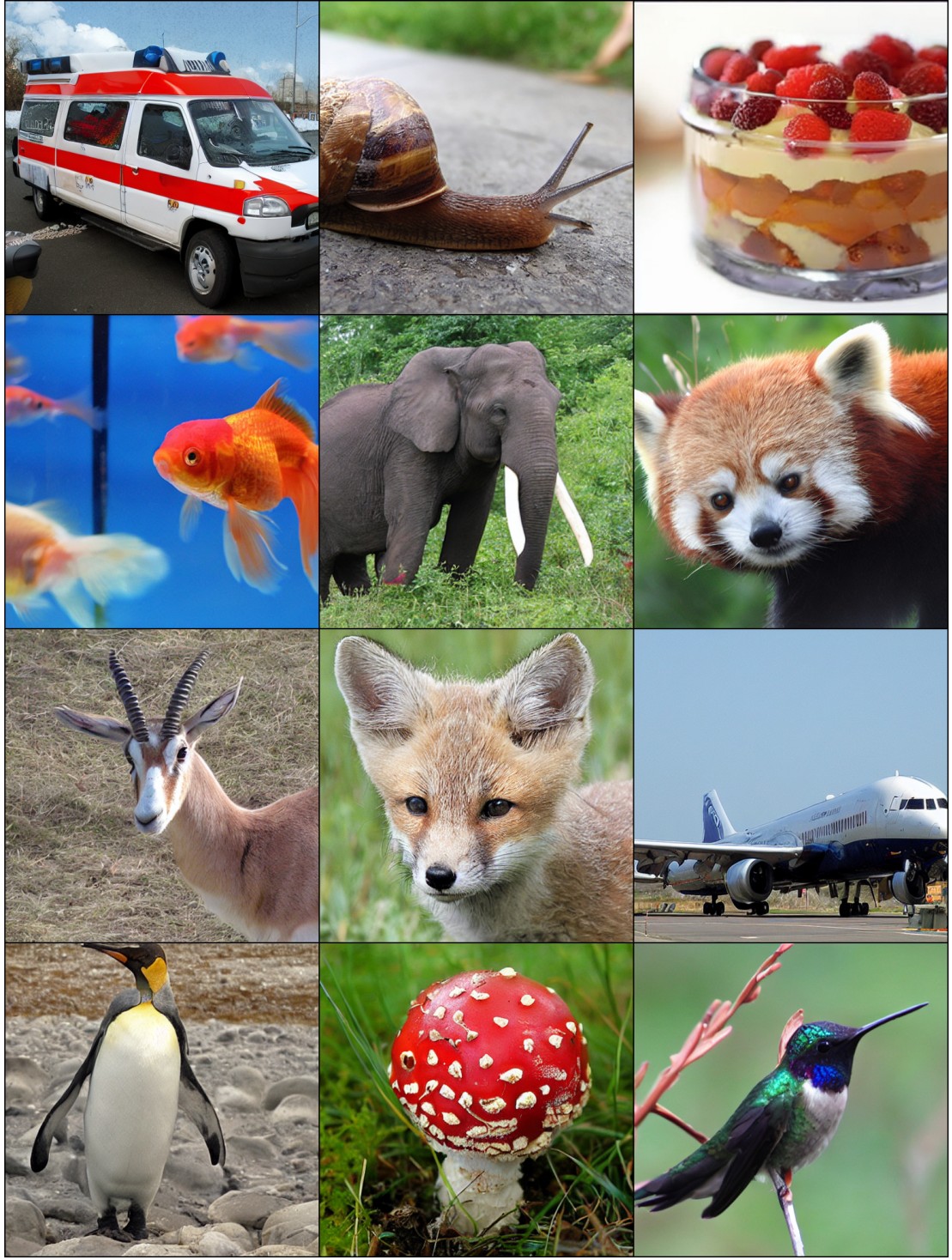

Figure 12: Class-conditional 512×512 image generation from MaskDiT with guidance 1.5 and 40 deterministic EDM (Karras et al., 2022) sampling steps.

