# OpenReview forum: "Fast Training of Diffusion Models with Masked Transformers"
_TMLR — Accepted by TMLR_

### Review · Reviewer_FZfH · 2023-11-16

**Summary Of Contributions:**

In this work, the authors proposed MaskDiT to accelerate the training speed of diffusion models, following the asymmetric encoder-decoder architecture. For the experiments on ImageNet-256$\times$256, MaskDiT largely reduces the computational cost without significant loss on performance compared with existing diffusion models.

**Audience:**

Yes

**Claims And Evidence:**

Yes

**Requested Changes:**

See the weaknesses.\
In general, the experiments really should be extended and articulation of the related work also needs to be improved.

**Strengths And Weaknesses:**

**Strengths**\
        **a**. The proposed MaskDiT largely saves the number of parameters to train diffusion models using the asymmetric encoder-decoder masked DiT as the backbone of diffusion models and the MAE loss.\
        **b**. The presentation is in general clear and comprehensive. The whole paper is easy to follow.\

**Weaknesses**\
        **a**. The experiments are limited to ImageNet-256$\times$256, where the commonly applied benchmark datasets for diffusion models with higher resolutions are not tested. Therefore, the main contribution of this work on the training efficiency should be further evaluated. The commonly applied high-resolution datasets for diffusion models include the CelebA-HQ dataset consisting of 30,000 images at $1024\times 1024$ resolution.\
        **b**. A minor weakness is that the motivation should be more coherent. There have been a growing number of pretrained diffusion models which are publicly available. Given these pretrained models, the sampling efficiency is the key to the efficiency of the diffusion models. To enhance the sampling efficiency, there have been a number of effective methods (Song et al., 2021; Lu et al., 2022; and Song et al., 2023). Therefore, why is it still necessary to  propose MaskDiT for higher the higher training efficiency?\
        **c**. Fast training and sampling of diffusion models has been also studied in other existing work including what follows the theoretical insights from optimization or advanced numerical solvers (Wu et al., 2023). However, the authors claimed that they cannot be directly applied to reduce the training cost of diffusion models. It would enhance the comprehensiveness of this work to provide a discussion on the differences between the other existing method (Wu et al., 2023) and MaskDiT.

---

### Review · Reviewer_eJnL · 2023-12-04

**Summary Of Contributions:**

The authors propose an architectural change and a special loss function to speed up training for diffusion models. The method appears to work according to well-established metrics of generative performance of image data.

**Audience:**

Yes

**Claims And Evidence:**

Yes

**Requested Changes:**

Please fix the following mistakes.

- "our approache achieves ~~the~~ similar performance ..." (Abstract)
- "Our main hypothesis is that ~~image~~ images contain* significant redundancy in the pixel space." (Introduction)
- "We propose a new approach for ~~the~~ fast training of ..." (Introduction)
- "trades a slightly ~~more~~ higher training cost for ~~the~~ better performance of ~~the~~ CFG sampling." (3.2 Key designs)
- Replace upper/bottom with left/right in the caption of fig 5

Also, you only have to cite Dosovitskiy once per section per mentioning of ViT.

**Strengths And Weaknesses:**

The claim of improving training speed is very relevant and well supported by the experiments.
The paper is well written except for some minor spelling and grammar mistakes (see list below of those I caught).

My major concern is that of using a loss function without a principled grounding in the marginal likelihood (such as a variational bound) and using it directly on a set of heuristics, like FID and IS. My personal experience with training such models is limited, so I have very little intuition on how often such models and their learning rules can escape such metrics and game them.
From what I know, randomly leaving out a reconstruction terms can be seen as an unbiased estimate of that reconstruction. Leaving out inputs/conditions however can lead to bias in the context of variational inference. A theoretical analysis would have been a great plus.

Still, I think that the paper and the experiments are executed well.

---

### Review · Reviewer_difn · 2024-01-03

**Summary Of Contributions:**

Thea authors propose a new faster training paradigm for transformer-based diffusion models which retains generation performance of the original model (Diffusion Transformer, DiT) while taking only 31% of its training time. The approach involves randomly masking out input patches, and only performing denoising score matching on unmasked tokens. This significantly reduces the memory and compute required to train on full tokens thus achieving faster training of these models. The intuition behind using masked tokens is that due to high visual redundancy in images, learning to denoise only a small set of unmasked patches in the image doesn't affect the generation performance. To prevent overfitting on unmasked patches and to understand global visual information, they add the MAE objective of reconstructing the masked patches via MSE loss as an auxiliary loss. The paper show experiments on class-conditional ImageNet-256x256 generation benchmark.

**Audience:**

Yes

**Claims And Evidence:**

Yes

**Requested Changes:**

See Weaknesses.

**Strengths And Weaknesses:**

Strengths:
- The authors propose a simple technique to improve training efficiency of training diffusion transformer models. The approach is motivated by  the same intuitions as the Masked Auto Encoder paper, and benefits from the same simplicity. I like that the approach is simple to implement, and is quite effective.
- I enjoyed reading the paper! It's very easy to follow, and the experiments section is fairly exhaustive. The ablations in the paper are well done and insightful.

Weaknesses:
Given the scope of the paper (improving diffusion transformer model training efficiency while maintaining generation performance), the paper claims are well supported by the experiments. However, there is one additional experiment that I'd like to see:
- I am curious to see if the proposed approach can be effective for higher resolution images. To complete the manuscript, it would be nice to show comparison on class-conditional ImageNet 512x512 with DiT.

---

### Comment · Action_Editor_JuCj · 2024-01-04
**Start of discussion period**

Dear authors,

Thank you for your submission to TMLR! Three reviews of your submission are now available for you to read. Please take this opportunity to answer the reviewers' questions and address their concerns. Your discussion with the reviewers will be instrumental in enabling a fully informed decision on this submission from the reviewers two weeks from now.

---

### Decision · Action_Editor_JuCj · 2024-02-08

**Recommendation:** Accept as is

**Comment:**

As acknowledged by the reviewers, the claims are empirically supported and the paper is easy to follow. Moreover, the authors improved their submission using the reviewers' feedback.

**Audience:**

This paper would be of interest for anyone interested in generative modelling (especially for those interested in diffusion models) and the masked transformers models beyond representation learning.

**Claims And Evidence:**

The claims of the submission concerning faster training are supported by the experiments presented in the paper. The claims, methods, and results are clearly explained and solid.